# Clinical characterization of Lassa fever: A systematic review of clinical reports and research to inform clinical trial design

**Laura Merson**[1,2], **Josephine Bourner**[1]*, **Sulaiman Jalloh**[3], **Astrid Erber**[1,4], **Alex Paddy Salam**[1], **Antoine Flahault**[2], **Piero L. Olliaro**[1]

1 Centre for Tropical Medicine and Global Health, Nuffield Department of Medicine, University of Oxford, Oxford, United Kingdom, 2 Institute of Global Health, Faculty of Medicine, University of Geneva, Geneva, Switzerland, 3 Ola During Children's Hospital, Freetown, Sierra Leone, 4 Department of Epidemiology, Center for Public Health, Medical University of Vienna, Vienna, Austria

* josephine.bourner@ndm.ox.ac.uk

**Data Availability Statement:** All relevant data are within the manuscript and its Supporting Information files.

## Abstract

### Background

Research is urgently needed to reduce the morbidity and mortality of Lassa fever (LF), including clinical trials to test new therapies and to verify the efficacy and safety of the only current treatment recommendation, ribavirin, which has a weak clinical evidence base. To help establish a basis for the development of an adaptable, standardised clinical trial methodology, we conducted a systematic review to identify the clinical characteristics and outcomes of LF and describe how LF has historically been defined and assessed in the scientific literature.

### Methodology

Primary clinical studies and reports of patients with suspected and confirmed diagnosis of LF published in the peer-reviewed literature before 15 April 2021 were included. Publications were selected following a two-stage screening of abstracts, then full-texts, by two independent reviewers at each stage. Data were extracted, verified, and summarised using descriptive statistics.

### Results

147 publications were included, primarily case reports (36%), case series (28%), and cohort studies (20%); only 2 quasi-randomised studies (1%) were found. Data are mostly from Nigeria (52% of individuals, 41% of publications) and Sierra Leone (42% of individuals, 31% of publications).

The results corroborate the World Health Organisation characterisation of LF presentation. However, a broader spectrum of presenting symptoms is evident, such as gastrointestinal illness and other nervous system and musculoskeletal disorders that are not commonly included as indicators of LF.

**Funding:** This work was supported by the UK Foreign, Commonwealth and Development Office and Wellcome [215091/Z/18/Z] (PO, JB) and the Bill & Melinda Gates Foundation [OPP1209135] (PO, JB). This project is part of the EDCTP2 programme supported by the European Union [RIA2016E-1612 - African coaLition for Epidemic Research, Response and Training (ALERRT)] (LM, SJ). For the purpose of Open Access, the author has applied a CC BY public copyright licence to any Author Accepted Manuscript version arising from this submission. The funders had no role in study design, data collection and analysis, decision to publish, or preparation of the manuscript.

**Competing interests:** The authors have declared that no competing interests exist.

The overall case fatality ratio was 30% in laboratory-confirmed cases (1896/6373 reported in 109 publications).

## Conclusion

Systematic review is an important tool in the clinical characterisation of diseases with limited publications. The results herein provide a more complete understanding of the spectrum of disease which is relevant to clinical trial design. This review demonstrates the need for coordination across the LF research community to generate harmonised research methods that can contribute to building a strong evidence base for new treatments and foster confidence in their integration into clinical care.

### Author summary

Clinical research in difficult-to-study infectious diseases such as Lassa fever is challenging. Only one controlled clinical trial has been conducted to assess the safety and efficacy of therapeutic interventions for Lassa fever (LF). Further research to test new and repurposed therapies is needed and should be supported by a methodological framework in which clinical trials can be consistently conducted for LF. To establish a basis for a standardised clinical trial methodology, we carried out a systematic review to identify the clinical characteristics and outcomes of LF and describe how LF has historically been defined and assessed in the scientific literature. Our data corroborates the current characterisation of LF, and also highlights a broader range of other general symptoms that characterise the onset of LF, such as gastrointestinal illness and other nervous system and musculoskeletal disorders that are not commonly included as indicators of LF. These findings, however, should be tempered by the lack of systematic assessment and reporting of presenting signs and symptoms, their evolution following treatment, and outcomes at discharge in the historic literature. It is therefore evident that a standardised set of data variables and outcome measures should be developed and incorporated into future trials and reported to accelerate collective knowledge.

## Introduction

Lassa Fever (LF) is an acute viral haemorrhagic disease caused by the Lassa virus, which is endemic to parts of West Africa, including Nigeria, Sierra Leone, Guinea and Liberia. Transmission to humans usually occurs through contact with excreta of infected rodents (primarily the Mastomys rat), [1,2] consumption or handling of contaminated food or household items (mostly involving women and children), or through direct contact with the bodily fluid of an infected person (typically healthcare workers) [1,3].

LF is a seasonal disease that is estimated to cause 100,000 to 300,000 new cases and 5,000 deaths each year. [4] The reported case fatality rate (CFR) is approximately 30% in patients who present to health care settings [5]–although this figure is lower (12%) in the most recent large cohort study taking place in a research setting. [6] During the peak season it is possible for large outbreaks to occur. Nigeria has experienced increases of confirmed cases and deaths every year since 2017, although this may be attributed to heightened clinical awareness and improvements in diagnostic capacity. [7–10]

Healthcare workers and pregnant women are considered to be at significant risk of severe Lassa fever outcomes. [11] High prevalence of LF has been identified in healthcare workers [5] which is thought to be a result of low levels of clinical suspicion of LF and an inadequate supply of quality protective equipment, making adherence to infection prevention and control (IPC) measures challenging. [12,13] Pregnant women are three-times more likely to have a fatal outcome than non-pregnant adults. [14]

No drug has so far received regulatory approval for treating LF. Ribavirin, in conjunction with supportive care, is currently used as the primary treatment for LF and has been incorporated into national and international treatment guidelines. [2,15] Ribavirin is on the World Health Organisation (WHO) list of essential medicines for treating viral haemorrhagic fevers. [16] However, this treatment recommendation has a small evidence base, as only a single clinical trial has been conducted to evaluate its effectiveness. [17,18] Further evidence in the form of clinical trials is required both to confirm the efficacy and safety of ribavirin and to test new therapies.

Multiple reasons conspire to make LF a neglected, difficult-to-study infectious disease of poverty. The narrow geographical spread of LF, which translates into a relatively limited number of patients who can be treated and studied in the few available specialised healthcare facilities, may contribute to low commercial interest for pharmaceutical companies. Consequently, a significant challenge to building the clinical evidence base is the lack of a methodological framework in which LF clinical trials can be reliably conducted in a consistent and comparable manner.

To establish a basis for the development of a standardised clinical trial methodology, we conducted a systematic review to identify the clinical characteristics and outcomes of LF, and to understand how LF has historically been defined and assessed in the scientific literature. The clinical characterisation enabled by this review is examined in combination with the diagnostics and demographics across the literature to expose the full spectrum of the disease that should be considered in optimising clinical trial methods.

This approach has previously been used to establish a foundation for the development of harmonised clinical trial methodologies, and specifically for the development of Core Outcome Sets, for other infectious diseases. For example, two systematic reviews of cutaneous leishmaniasis interventions conducted by Gonzalez and colleagues (2008 & 2009) [19,20] formed the knowledge-base for standardised trial methodology proposed by Olliaro and colleagues (2013)[21], and a review of tuberculosis meningitis research informed the definition of standardised diagnostic criteria, [22] data collection and outcomes for future clinical trials.[23]

## Methodology

An initial search of Epistemonikos and Prospero was conducted to understand if any high-quality reviews were available or in progress that covered portions of the planned review.

For the main search, the following databases and clinical trial registries were searched for clinical studies: African Journals Online, Cochrane Central Register of Controlled Trials (CENTRAL), Embase, Global Health (Ovid), Global Index Medicus, PubMed/MEDLINE, clinicaltrials.gov, ISRCTN, Pan African Clinical Trials Registry, and WHO International Clinical Trials Registry. Cochrane Database of Systematic Reviews, Cochrane Clinical Answers, and COMET databases were also searched (**S1 Text**). Studies of LF published before 15 April 2021 were included with no language restrictions applied. We registered this review in PROSPERO, the international prospective register of systematic reviews of the University of York and the National Institute for Health Research, under protocol number CRD42020220365. [24]

We included only studies reporting primary results of patients with a laboratory or clinically confirmed diagnosis of LF (including all diagnostic methods and studies that report patients with sequelae from a previous infection) and describing LF clinical features and/or treatments.

All identified studies underwent screening for inclusion/exclusion in two stages: first by review of titles and abstracts, then by review of the full text manuscripts. Screening was conducted in Rayyan, [25] aided by pre-defined inclusion/exclusion key words. At both stages of the review, decisions on the inclusion or exclusion of each study required the agreement of two unique reviewers assessing the reference or manuscript independently. Each reviewer's decisions were blinded to the second reviewer. When a dataset was fully screened by both reviewers, decisions were unblinded and conflicts were resolved by discussion between the two reviewers, or with a third reviewer when required, to agree on inclusion or exclusion.

Data were extracted from the selected manuscripts and entered into a Research Electronic Data Capture (REDCap) database using a predefined variable dictionary available in **S1 Table**. [26] A second reviewer verified all extracted data and resolved any discrepant data by reviewing the manuscript with a third reviewer. During full-text screening, the most frequently reported signs and symptoms of interest were identified by the review team. For each publication, the presence or absence of information on each of these 19 pre-defined signs and symptoms was recorded where available. Several other signs and symptoms of LF were also reported. Data on these characteristics were extracted and assessed for clinical significance by clinicians in the study team. Signs and symptoms considered to be clinically significant are reported alongside those that were pre-identified. All other reported signs and symptoms can be found in **S2 Table**.

For the sections in this systematic review that report data relating to case fatality, signs and symptoms of LF, data are presented only for studies that include populations with 100% laboratory confirmation (N = 122)–all other sections in this review report data for all publications (N = 147) regardless of the proportion of the study population with laboratory confirmation of LF.

The prevalence of signs and symptoms is reported based on the number of patients for whom there was evidence of assessment of each sign or symptom in the publication (number of patients with sign or symptom reported/over number of patients assessed). Results relating to bleeding site are reported as a percentage of publications that include both location of bleeding site and number (or proportion) of individuals exhibiting bleeding at specified sites.

The minimum and maximum time in days were extracted for data relating to time from symptom onset to presentation and is reported alongside the median, interquartile range (IQR) and range. For data relating to time from presentation to death, the median of reported mean times in days, the IQR and the range are reported. Aspartate aminotransferase (AST) is reported categorically as $<150$ IU/L and $\geq150$ IU/L to understand liver enzyme levels in the context of the evidence base that has informed current treatment guidelines. [17]

Risk of bias assessments were conducted using the Joanna Briggs Institute (JBI) critical appraisal tools due to the compatibility of these tools with the variety of study types in this review. [27,28] Tools were applied to each study based on criteria outlined in the JBI Manual for Evidence Synthesis. [29] Diagnostic methods were evaluated for confidence of acute LF diagnosis based on acceptance in the existing literature and the results of recent reviews. [30,31]

## Results

### Search results

In total, 4,794 publications were identified in the literature search. After removing duplicates, 2,704 titles and abstracts were screened and 195 full-text publications were assessed for eligibility, resulting in the inclusion of 147 publications in the data synthesis (**Fig 1**).

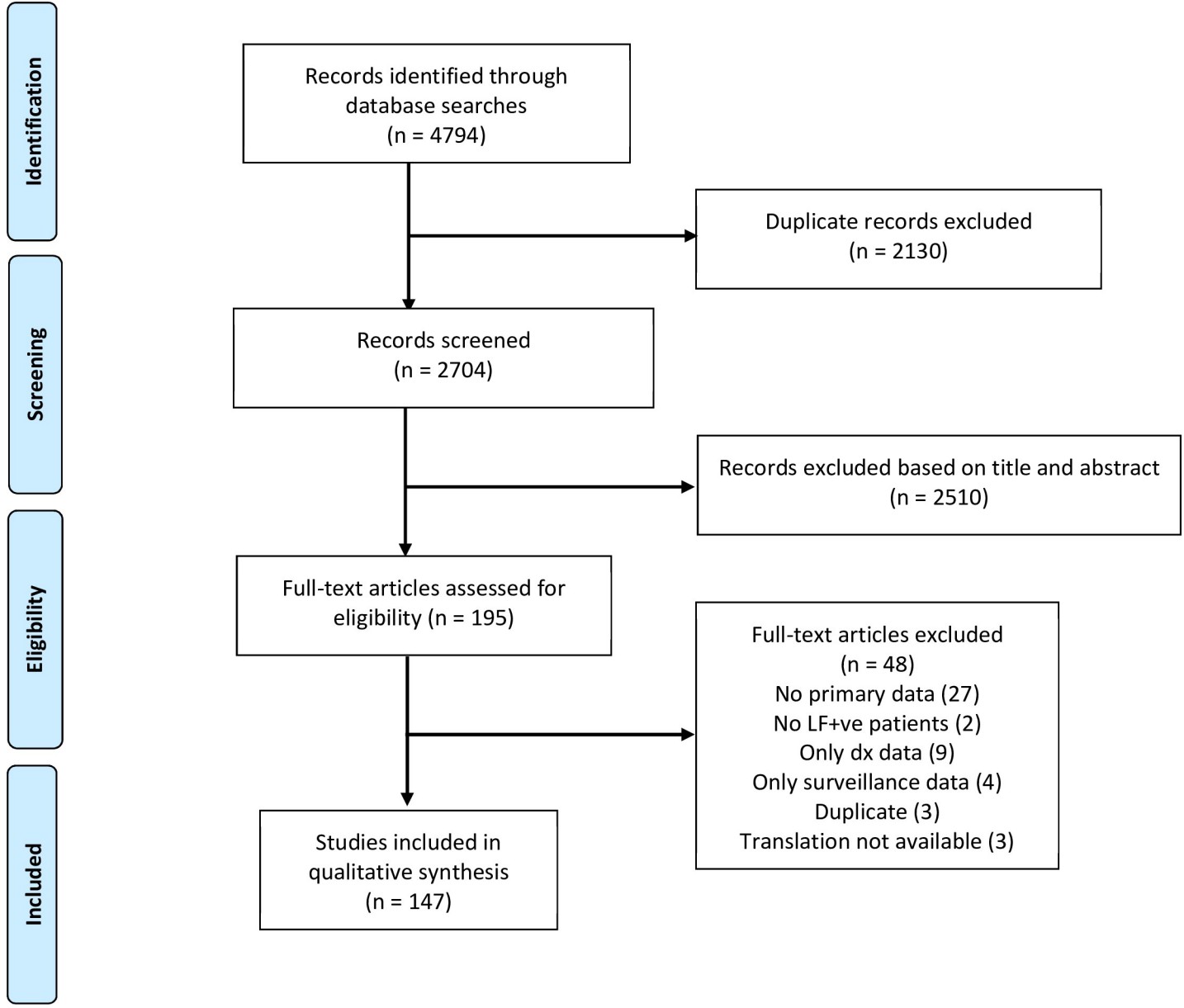

**Fig 1. PRISMA 2009 flow diagram.**

Of the included publications, 53 (36%) were case reports, 41 (28%) were case series, 30 (20%) were cohort studies, 10 (7%) were case-control studies, 11 (7%) were cross sectional studies, and 2 (1%) were quasi-randomised studies (**Table 1** and **S2 Text**).

The publication years ranged from 1970 to 2021 with increasingly higher numbers of publications being generated after 2010 (**Fig 2**).

## Risk of bias assessments

Overall, there was a moderate risk of bias across all outcomes evaluated in the risk of bias assessment (**Fig 3**). An average of 62% (IQR 44–83%) of study-specific criteria in the JBI Critical Appraisal Checklists were included in the publications. The criteria most relevant to this review

**Table 1. Number of studies included by study type.**

| | |
|---|---|
| Total publications (N) | 147 |
| Publication type | N (%) |
| Full text publication | 132 (89.7) |
| Conference abstract | 10 (6.8) |
| Letter/short communication | 5 (3.4) |
| Study type[1] | N (%) [2] |
| Case report(s) | 53 (36) *[32–85]* |
| Case series | 41 (28) *[86–125]* |
| Cohort study | 30 (20) *[6,126–154]* |
| Case-control study | 10 (7) *[155–164]* |
| Cross-sectional study | 11 (7) *[165–175]* |
| Quasi-randomised | 2 (1) *[17,176]* |

[1]Definitions have been adapted from the Cochrane Community Glossary, available at community.cochrane.org/glossary (**S2 Text**)

[2] Citations shown in bold italics

were often missing or unclearly reported including: Inclusion criteria clearly defined in 51% of 47 publications, Demographic characteristics described in 34% of 88 publications, Exposure measured reliably in 62% of 109 publications, and Adverse events reported in 53% of 60 publications. These gaps in reporting may result in an incomplete characterisation of LF throughout the clinical course of disease. Though an appropriate statistical analysis was reported in 96% of 79 publications, strategies to deal with confounders were reported in only 19% of 79 publications.

Furthermore, 74 (50%) publications are case reports or case series describing fewer than 6 individuals. The basis for the selection of the enrolled participants is often unclear and may not be representative of the patient population who present to the health centre, meaning generalising the findings of these studies to the wider LF patient population is not feasible.

## Population characteristics

In total, data on 8550 individuals were reported in the selected publications (**Table 2**). Most individuals (91%) in the reported population were enrolled in studies conducted in either Sierra Leone or Nigeria. The remainder of the study population were enrolled in studies conducted in West Africa, Europe, Asia and North America (**Table 2**).

Note that reporting details do not enable the delineation of overlapping patient populations reported across multiple publications. Therefore, some patients may have been included in more than one publication.

Of the individuals included in the publications where sex was noted, 3477 individuals (48%) were male and 3839 were female (52%). Sex was not reported for 1234 individuals (14%).

Of the publications in which the age range of patients was reported (N = 129), 48 (37%) included a patient under the age of 16 years. 35 publications (24%) reported the inclusion of at least one pregnant participant.

## Eligibility criteria

68 publications (46%) documented eligibility criteria (**Table 3**). Of these, all 68 (100%) reported inclusion criteria and 5 (7%) reported exclusion criteria.

Within the inclusion criteria, 20 publications (29%) specified signs and symptoms that must be present for inclusion, 17 (25%) specified fever should be present upon enrolment. 27

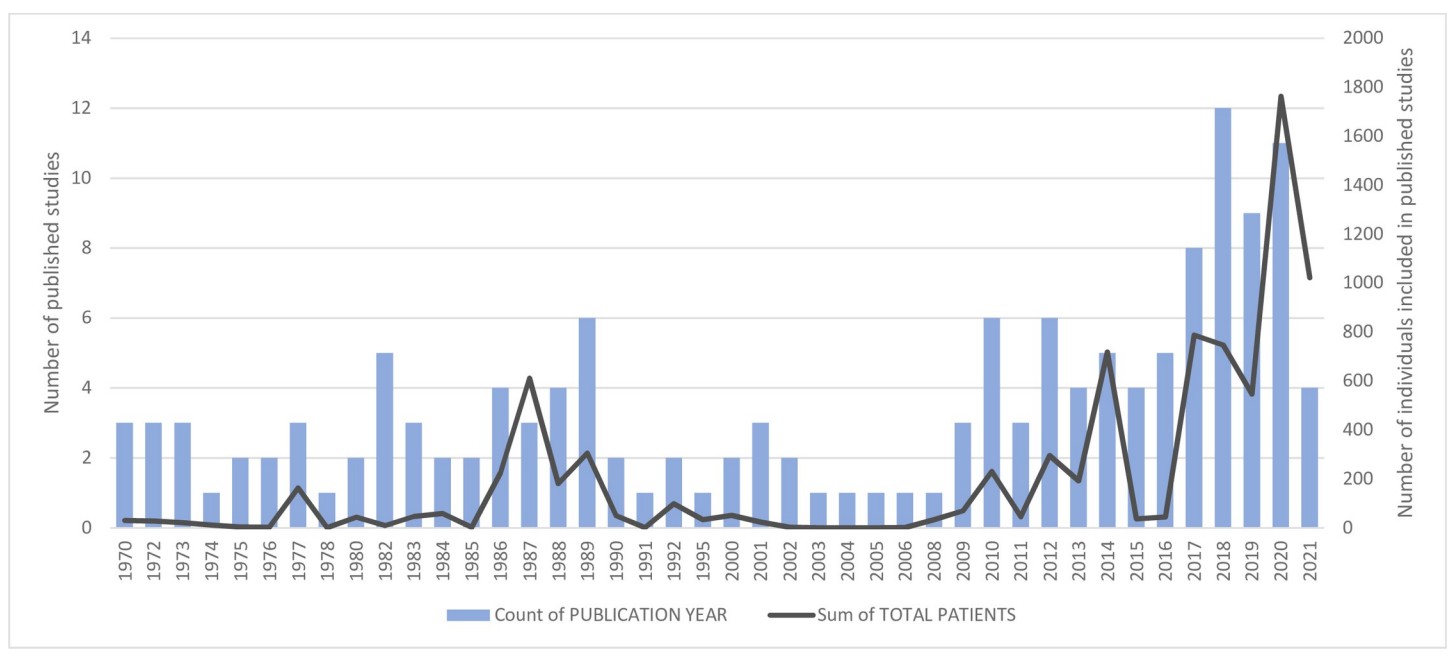

**Fig 2. Number of Lassa fever clinical studies published per year.**

publications (40%) required laboratory confirmation of LF either alone or in conjunction with clinical symptoms. 26 publications (38%) required a confirmation or diagnosis of LF without specifying a method of diagnosis.

Exclusion criteria were based on geographic location in 2 publications (3%). Hearing loss, co-infection with Marburg virus, contacts of an index LF case, patients with an incomplete data set, and LF patients managed on an out-patient basis were each an exclusion criteria in 1 publication (1%).

## Outcome measures

Six publications (4%) defined a primary endpoint or outcome measure, 5 (83%) of which were cohort studies and 1 (17%) of which was a quasi-randomised trial. Five (83%) publications included mortality as the primary outcome measure and one (17%) included diagnosis of acute kidney injury (AKI).

Three publications (2%) reported at least one secondary endpoint or outcome measure. Viral load throughout treatment was an outcome measure in 2 publications (66%). Aspartate aminotransferase (AST) throughout treatment, live birth, all-cause in-hospital fatality, frequency of acute kidney dysfunction, prognosis of AKI and AKF in terms of estimated glomerular filtration rate (eGFR) at the end of follow-up and time to hospital discharge were each listed as an outcome measure in 1 publication.

The period of follow-up was specified in 97 publications (66%). The median follow-up time was 21 days (IQR 9–67) with a range of 1–10,950 days.

## Method of case confirmation

Laboratory-confirmed diagnosis was the most prevalent method of case confirmation, used either alone or in conjunction clinical diagnosis in 141 publications (96%) and 8422 individuals (99%). Two publications (1%) confirmed LF using clinical diagnosis alone and 4 publications (3%) did not specify a method of case confirmation.

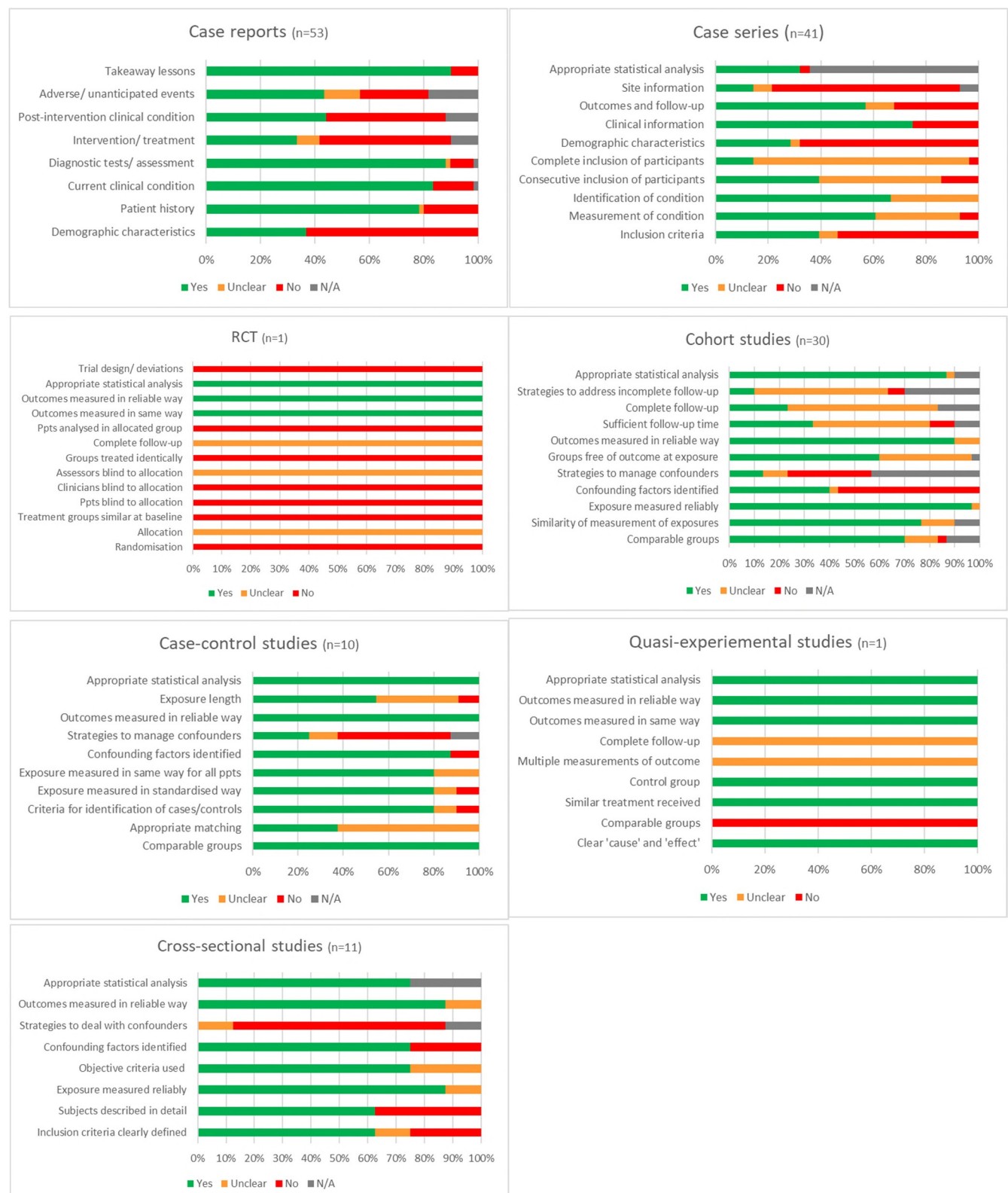

**Fig 3. Risk of bias assessment summary by outcome and study type.**

**Table 2. Individuals and publications included in the systematic review per country.**

| Country | N individuals (%) | N publications (%)[1,2] |
|---|---|---|
| Nigeria | 4461 (52) | 61 (41) [6,32,33,37,45,47,54,55,58,68,70,81,84,86–90,93,94,97,99,100,104,109–113,119,121,123,124,126–128,133–135,138,140,141,145,149–152,155–157,159,164,171–173,175] |
| Sierra Leone | 3563 (42) | 45 (31) [17,42,46,50,53,67,76,77,82,85,93,95,96,107,118,120,122,130–132,139,142–144,146,148,153,154,158,160–163, 165,169,174,176,177] |
| Liberia | 373 (4) | 10 (7) [35,36,69,80,105,106,108,136,148,170] |
| Benin | 73 (1) | 2 (1) [39,125] |
| Guinea | 30 (<1) | 3 (2) [83,122,129] |
| United States | 16 (<1) | 13 (9) [38,43,44,57,60,62,66,73,91,93,103,116,147] |
| UK | 10 (<1) | 8 (5) [41,48,49,59,65,78,101,115] |
| Ghana | 6 (<1) | 3 (2) [92,98,102] |
| Germany | 4 (<1) | 4 (3) [51,61,116,117] |
| Netherlands | 4 (<1) | 4 (3) [72,74,75,117] |
| Canada | 2 (<1) | 2 (1) [40,64] |
| Japan | 2 (<1) | 2 (1) [56,79] |
| Togo | 2 (<1) | 1 (1) [114] |
| Ireland | 1 (<1) | 1 (1) [63] |
| Israel | 1 (<1) | 1 (1) [178] |
| Ivory Coast | 1 (<1) | 1 (1) [34] |
| Sweden | 1 (<1) | 1 (1) [52] |

[1]N >147 and % >100 as some publications include individuals from >1 country

[2]Citations shown in bold italics

The largest proportion of laboratory diagnoses, for 2348 individuals (27%), was conducted using RT-PCR alone (**Table 4**. Results are shaded according to the level confidence of laboratory methods for the identification of acute LF infection.)

When RT-PCR, viral culture and antigen ELISA were used exclusively or in combination with each other, we determined a high level of confidence of acute LF infection. A moderate level of confidence was determined when serology (IgM ELISA and/or IgG ELISA) was used in combination with RT-PCR, viral culture or antigen ELISA. Any other exclusive or combined use of laboratory methods was determined as a lower confidence of acute LF infection.

## Baseline clinical characteristics

At baseline, fever was the most reported symptom, identified in 88% of the individuals in whom it was assessed (1527/1730) (**Fig 4**), followed by headache (809/1622 individuals, 50%), vomiting (806/1613, 49%), abdominal pain (660/1581, 42%) and cough (556/1581, 35%).

**Table 3. Specified inclusion criteria by type (N = number of studies).**

| Criteria | N (%) |
|---|---|
| Publications specifying eligibility criteria (inclusion or exclusion) | 68/147 (46) |
| Publications specifying inclusion criteria | 68/68 (100) |
| Laboratory confirmation of LF either alone or in conjunction with assessment of clinical symptoms | 27/68 (40) |
| Confirmation or diagnosis of LF (method of diagnosis unspecified) | 26/68 (38) |
| Patients with pre-specified signs and symptoms consistent with LF | 20/68 (29) |
| Patients with fever alone or in conjunction with other symptoms | 17/68 (25) |
| Publications specifying exclusion criteria | 5/68 (7) |
| Geographic location | 2/5 (40) |
| Hearing loss (sensorineural or conductive hearing loss) | 1/5 (20) |
| Co-infection with Marburg virus | 1/5 (20) |
| Contacts of index LF case | 1/5 (20) |
| Patients managed on an out-patient basis | 1/5 (20) |
| Patients with an incomplete dataset | 1/5 (20) |

A smaller number of patients presented with clinically severe or life-threatening signs and symptoms such as shock (12/187, 6%), breathing difficulty (21/310, 7%), and seizure (13/517, 3%). In pregnant women, labour complications were reported in 2/7 (29%).

Overall, musculoskeletal disorders, nervous system disorders and gastrointestinal illnesses were most reported. (**S3 Table**).

The reported timeframe from symptom onset to presentation ranged from 0–32 days (median reported minimum time: 5 days; median reported maximum time: 8 days). The minimum and maximum times to presentation were similar in Nigeria, Sierra Leone and Liberia, the three endemic countries with >5 studies included.

## Post-baseline clinical characteristics

In post-baseline assessment, fever was again the most prevalent symptom, reported for 3067/3300 individuals (93%)–representing a 5% increase in prevalence from baseline.

After fever, headache (2033/3200, 64%), vomiting (1695/3077, 55%) and abdominal pain (1594/3039, 52%) occurred in the highest number of individuals. The prevalence of all signs and symptoms related to the gastrointestinal system increased post-baseline, except nausea.

The number and prevalence of reported severe or life-threatening signs and symptoms also increased post-baseline. Shock had the greatest increase in prevalence from baseline to post-baseline, followed by breathing difficulty and seizure. Respiratory failure and renal failure were reported only at post-baseline timepoints.

1896/6373 (30%) individuals were reported to have died. 109 publications (74%) reported at least one death. 41 publications (31%) reported the time in days from presentation to death and within these publications, the median of the mean times reported per publication from presentation to death was 7.5 days (IQR 3–11) with a range of 0–21 days.

**Site of bleeding.** Bleeding was reported in 30 (20%) and 53 (36%) publications at baseline and post-baseline, respectively. At baseline, site of bleeding was reported for 168 individuals and at post-baseline site of bleeding was reported for 1088 individuals (**Table 5**).

Haematuria/blood in urine was the most reported bleeding type at both baseline and post-baseline–reported for 23 individuals (14% of reported bleeding sites) at baseline and 131 individuals (12% of reported bleeding sites) at post-baseline. Haematuria is specified as macroscopic haematuria in a third of the reported cases, while no differentiation between macroscopic and microscopic haematuria was made in two thirds of these cases.

**Table 4. Laboratory Diagnostics—Single and combined testing methods by number of individuals tested.**

| | Included N (%)* | Exclusively N (%) | RT-PCR N (%) | Viral Culture N (%) | IFA N (%) | IgM ELISA N (%) | Antigen ELISA N (%) | IgG ELISA N (%) | CF N (%) | IHC N (%) | Other N (%) |
|---|---|---|---|---|---|---|---|---|---|---|---|
| RT-PCR | 4127 (48) | 2348 (27) | ■ | 1497 (18) | 79 (<1) | 1549 (18) | 268 (3) | 156 (2) | 0 (0) | 1355 (16) | 46 (<1) |
| Viral Culture | 3192 (37) | 28 (<1) | 1497 (18) | ■ | 1638 (19) | 1566 (18) | 73 (<1) | 213 (2) | 118 (1) | 1355 (16) | 155 (2) |
| IFA | 2232 (26) | 99 (1) | 79 (<1) | 1638 (19) | ■ | 497 (6) | 361 (4) | 188 (2) | 217 (3) | 0 (0) | 155 (2) |
| IgM ELISA | 3542 (41) | 0 (0) | 1549 (18) | 1566 (18) | 497 (6) | ■ | 2009 (23) | 911 (11) | 0 (0) | 1354 (16) | 642 (8) |
| Antigen ELISA | 2204 (26) | 78 (<1) | 268 (3) | 73 (<1) | 361 (4) | 2009 (23) | ■ | 775 (9) | 0 (0) | 1 (<1) | 643 (8) |
| IgG ELISA | 919 (11) | 4 (<1) | 156 (2) | 213 (2) | 188 (2) | 911 (11) | 775 (9) | ■ | 0 (0) | 1 (<1) | 643 (8) |
| CF | 307 (4) | 30 (<1) | 0 (0) | 118 (1) | 217 (3) | 0 (0) | 0 (0) | 0 (0) | ■ | 1 (<1) | 0 (0) |
| IHC | 1357 (16) | 0 (0) | 1355 (16) | 1355 (16) | 0 (0) | 1354 (16) | 1 (<1) | 1 (<1) | 1 (<1) | ■ | 0 (0) |
| Other** | 1187 (14) | 389 (5) | 46 (<1) | 155 (2) | 155 (2) | 642 (8) | 643 (8) | 643 (8) | 0 (0) | 0 (0) | ■ |

Green = high confidence of acute LF infection; Yellow = moderate confidence; Orange = low confidence.

Abbreviations: Reverse Transcription Polymerase Chain Reaction (RT-PCR); Immunofluoresence Assay (IFA); Immunoglobulin M (IgM); Immunoglobulin G (IgG); Complement Fixation (CF); Immunohistochemistry (IHC)

*Testing method not reported in 3 (2%) publications, including 5 (4%) individuals

** Other includes: Experimental LASV Antigen Rapid Test cassettes and dipstick LFI = 2 (2%) publications and 45 (1%) individuals; Lateral flow immunoassay (LFI) = 2 (2%) publications and 598 (10%) individuals; LF-specific antibody titre = 1 (1) publication and 154 (3%) individuals.

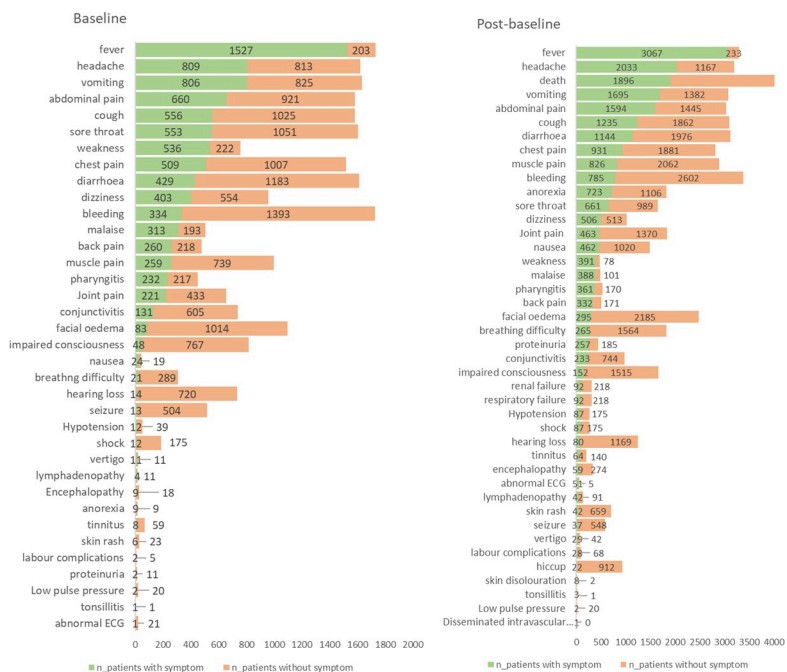

**Fig 4. Clinical signs and symptoms reported at baseline and post-baseline by proportion of individuals with and without sign/symptom.**

**Table 5. Number and percentage of bleeding sites reported.**

| BLEEDING SITES | Baseline n/N (%) | Post-baseline n/N (%) |
|---|---|---|
| (Sub) conjunctiva/eyes | 10/168 (6) | 12/1088 (1) |
| Ears | 0 | 1/1088 (<1) |
| Epistaxis (nose) | 8/168 (5) | 17/1088 (2) |
| Gingiva/mouth/buccal mucosa | 9/168 (5) | 20/1088 (2) |
| Haematemesis/vomit/upper GI | 14/168 (8) | 38/1088 (3) |
| Haematuria/urine | 23/168 (14) | 131/1088 (12) |
| Haemoptysis/pleural effusion/cough/respiratory/sputum | 1/168 (1) | 14/1088 (1) |
| Needle sites/ wounds | 2/168 (1) | 13/1088 (1) |
| Stool/rectum/melena/haematochezia/GI/overt | 3/168 (2) | 78/1088 (7) |
| Vagina | 1/168 (1) | 59/1088 (5) |

## Aspartate aminotransferase (AST)

Aspartate aminotransferase (AST) levels were reported at baseline and post-baseline in 29 (20%) and 43 (29%) publications respectively. These publications included 799 (9%) and 1200 (14%) of the populations described in the literature. Detailed, individual patient data were limited. Many publications reported a single AST value from a single patient, others presented a range and mean value for a group of patients. Of the publications that reported AST levels, 10 publications including a total of 19 (2%) individuals reported all patients as having baseline AST levels below 150 IU/L and 15 publications including a total of 459 (57%) of individuals reported AST values ≥150 IU/L for all individuals. AST values reported in other papers showed a wide range of results. (**Table 6**).

## Discussion

Despite fifty years of reporting and research, morbidity and mortality caused by LF remain high in patients presenting to health care settings, especially in high-risk populations such as healthcare workers and pregnant women. [11,14] A historic lack of investment in LF clinical research and LF drug development [179] has resulted in the limited anthology of case reports and observational study reports presented in this review, but no well-conducted randomised trials of current or new therapeutics.

Attention and investment in LF have increased in recent years with the launch of the World Health Organisation (WHO) Research & Development Blueprint [180] and the Coalition for Epidemic Preparedness Initiatives (CEPI) 'Enable' project. [181] Since 2018, LF is also included in the US Food and Drug Administration (FDA) list of "infectious disease for which there is no significant market in developed nations and that disproportionately affects poor and marginalized populations", which entitles the developer of an LF drug successfully registered with the US FDA to a priority review voucher (PRV). [182] These developments will hopefully promote new additions to the sparse evidence presented herein.

### Challenges and limitations of the current literature

Both the context and the presentation of LF present challenges to timely diagnosis and treatment, which are reflected in the limitations of the literature included in this review. (186–189) Resources, communications, and awareness of LF are reported to be increasing over time in the most affected countries, [183,184] though challenges to the effectiveness of these interventions are frequently met. [185,186] Differences in LF lineages circulating among the reported populations can also not be considered in this review as this information is not included in the

**Table 6. AST ranges in papers reporting AST values.**

|  | Total reporting AST values | | All AST values <150 IU/L | | All AST values ≥150 IU/L | |
|---|---|---|---|---|---|---|
|  | Publications N | Individuals n | Publications N (%) | Individuals n (%) | Publications N (%) | Individuals n (%) |
| AST baseline | 29 | 799 | 10 (34) | 19 (2) | 15 (52) | 459 (57) |
| AST anytime | 43 | 1279 | 9 (21) | 18 (1) | 14 (33) | 284 (22) |

clinical reports nor is clinical information detailed in publications on strain identification [187–189].

Due to these constraints, the limited quality of the literature, and the risk of bias demonstrated in the analysis, our report is strictly descriptive of LF clinical features and cannot inform the impact of LF treatment.

The purpose of this review is to ensure that future research, including the development of standardised data variables and outcomes for LF clinical trials, reflects the entirety of the available evidence base. Additionally, we aim to improve the uniformity of data collection and reporting in future LF research to support efficiency and comparison across studies.

## Case definition

The WHO estimates that 80% of cases are asymptomatic. It further defines common symptoms to include gradual onset of fever, malaise and general weakness; after a few days, extending to headache, sore throat, muscle pain, chest pain, nausea, vomiting, diarrhoea, cough, and abdominal pain; as well as possible bleeding, neck/facial swelling and shock in severe cases. [190] The Nigeria Centre for Disease Control (NCDC) suspected case definition is similar, but lists a more restricted set of indicative signs and symptoms of LF. [2] The literature reviewed reveals a much broader spectrum of symptoms than that of both the NCDC and WHO. Expanding the case definition may have advantages e.g., increasing the sensitivity of suspect case identification, the likelihood of rapid access to LF diagnostics, and better protecting healthcare works; however, it may also have disadvantages since these symptoms are non-specific and present in many common febrile illnesses across West Africa, [191,192] the reduced specificity of a case definition that includes them would need to be balanced by the availability of appropriate diagnostics to confirm suspect cases. In the context of clinical research, these issues are tempered by requiring laboratory confirmation for inclusion into the study.

Efforts are currently underway to improve the understanding of regional differences in LF presentation. Data from the CEPI-funded "Enable" project–a large multi-country epidemiological study across five countries in West Africa–aims to summarise the regional prevalence of LF and delineate key differences in its characterisation on a country-by-country basis.[181] These results may inform drivers of diversity in case definition.

## Diagnostic variation and confidence

11 different diagnostic approaches are used in the publications included in this review. This variety reflects the history of LF diagnostic development, resource limitations and research on new methods. When multiple tests are used to confirm LF diagnosis, the reported results do not specify which of the tests had a positive result. Furthermore, despite evidence that testing methods, kits and quality assurance protocols have a significant impact on diagnostic quality, [193–195] these details are rarely included in clinical reports and could therefore not be evaluated.

The US Centers for Disease Control and Prevention (CDC) guidelines recommend nucleic acid detection by RT-PCR, antigen detection by ELISA, or serology (IgG or IgM ELISA) for LF diagnosis. [196] The World Health Organisation guidelines concur and further add viral culture as acceptable. [197] In this review, results of studies that confirm LF diagnosis using RT-PCR, antigen ELISA and viral culture exclusively or in combination with each other are determined to have a high level of confidence in the diagnosis. Studies that accept IgM and/or IgG ELISA testing equally with these methods have a reduced confidence on the basis of evidence that detection of IgG and IgM can occur across a wide range of time points ranging from early in an infection and after resolution of acute illness. [198] Investment to improve the accuracy and availability of rapid diagnostics appropriate for LF endemic settings is needed.

## Need for more clinical trials

To date, a single clinical trial has been conducted to assess the safety and efficacy of ribavirin, the only recommended treatment for LF. [17] No further clinical trials for LF have been conducted in the subsequent 35 years. The analysis of this trial includes only a fraction of the cohort who received ribavirin and reports only those with an AST >150 IU/L. Reassessment of the results of this trial, and a subsequent report released under the Freedom of Information Act 2000, [199] identified harmful effects of using ribavirin to treat LF in patients with AST <150 IU/L. [18] Despite this significant limitation, ribavirin has been incorporated in to LF treatment guidelines without reference to AST-dependent dosing. [2]

This review cannot contribute much to knowledge of AST levels in LF, as they are minimally reported in the literature (15% of patients in this review). Where results are available, they are not sufficiently detailed to understand the distribution of AST values, possibly due to limited laboratory capacity. [200] In the reported data, publications including 459 (57%) patients reported all patients as having baseline AST levels ≥150 IU/L, above the limit of where harm from ribavirin has been identified. However, publications including only 284 (22%) patients reported all patients having AST levels ≥150 IU/L throughout admission. The potential for harm when AST levels are <150 IU/L after the start of ribavirin treatment has not been examined. Systematic assessment of AST and publication of comprehensive data needs to be undertaken to build understanding of the use of ribavirin in LF.

The shortcomings of ribavirin demonstrate an urgent need for its reassessment and highlight the need for other therapeutic options to be explored–particularly those that can be used in LF patients at heightened risk of death, including pregnant women. [14] There are a number of antivirals that are currently in development or under investigation for LF. [201] With data urgently needed on the safety and efficacy of these new treatment prospects, and in the interest of efficiency, it is vital that clinical trials are conducted and reported in a comparable manner. The spectrum of symptoms and sequalae described in this review provides focus to the targets of therapeutic action and safety considerations.

## The importance of standardising data collection and reporting

By collating and describing this history of results, we have shown that understanding of the prevalence of many signs and symptoms of LF is significantly limited by a lack of consistency in reporting of clinical features. There is a clear need for robust reporting of clinical studies and consensus in approaches to clinical characterisation that will allow for comparability of the characterisation of LF across regions and strains of the Lassa virus. The synthesises reported herein serves as a baseline for defining a standardised way to capture and report clinical characteristics across future LF research studies.

## Ways forward

Despite the growing number of research outputs on LF, the limited number of cases and the challenges of mounting clinical trials in LF-endemic regions highlight the need for efficient approaches to research. The results of this review have been leveraged to inform the development of standardised clinical trial methodologies with efficient and pragmatic design to address the research priorities above. A consultation group has been established to develop–through a consensus approach–clinical trial eligibility criteria, case definition, core data collection variables and outcomes. [202]

These criteria, definitions and measures can be integrated into pre-positioned protocols, available for future outbreaks. Pre-positioned protocols have been proposed and adopted for other emerging infections as a way of tackling the challenges of conducting research on sporadic diseases that require a rapid response. [203] Not only do pre-positioned protocols have the advantage of accelerating the pace at which research can start, but they also enable multiple studies to be conducted in a comparable way.

Our systematic review demonstrates the need to collect data on LF clinical characteristics and clinical management in a structured and harmonised fashion with defined core data requirements. This will lead to improvements in the understanding and clinical diagnosis of LF. It will further inform the design of clinical trials of existing and new treatments, and the potential application of indicators for pharmacological treatments, including the severity of disease requiring treatment. Coordinating high-quality research methods across the LF research community can contribute to building a strong evidence base on new treatments for all LF patients and foster confidence in their integration into clinical care.

## Supporting information

**S1 Text. Search Strategy.**
(DOCX)

**S2 Text. Study type definitions.**
(DOCX)

**S1 Table. Data dictionary and extraction manual.**
(DOCX)

**S2 Table. Other reported signs and symptoms.**
(DOCX)

**S3 Table. Prevalent and clinically significant signs and symptoms.**
(DOCX)

**S1 Data. Full raw data.**
(XLSX)

## Acknowledgments

The authors would like to thank Eli Harriss who conducted the search, Fernando Gouvea Reis who assisted with screening and data extraction, Maren Jeleff-Entscheff, Jared Palazza and Wahdae-Mai Harmon who assisted with screening, Andrew Dagens and Louise Sigfrid who contributed to data extraction, Kristen Eberhardt who provided results from the literature search for comparison and Jake Dunning for his critical review.

## Author Contributions

**Conceptualization:** Laura Merson, Astrid Erber, Piero L. Olliaro.

**Data curation:** Laura Merson, Sulaiman Jalloh.

**Formal analysis:** Laura Merson, Josephine Bourner.

**Funding acquisition:** Laura Merson.

**Methodology:** Laura Merson, Astrid Erber, Antoine Flahault, Piero L. Olliaro.

**Project administration:** Laura Merson, Josephine Bourner.

**Supervision:** Antoine Flahault, Piero L. Olliaro.

**Visualization:** Josephine Bourner.

**Writing – original draft:** Laura Merson, Josephine Bourner.

**Writing – review & editing:** Laura Merson, Josephine Bourner, Sulaiman Jalloh, Astrid Erber, Alex Paddy Salam, Piero L. Olliaro.

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
