## [Decision Letter · Decision Letter 0]

29 Jun 2021

Dear Mrs. Bourner,

Thank you very much for submitting your manuscript "Clinical characterization of Lassa fever: a systematic review of clinical reports and research to inform clinical trial design" for consideration at PLOS Neglected Tropical Diseases. As with all papers reviewed by the journal, your manuscript was reviewed by members of the editorial board and by several independent reviewers. In light of the reviews (below this email), we would like to invite the resubmission of a significantly-revised version that takes into account the reviewers' comments. 

We cannot make any decision about publication until we have seen the revised manuscript and your response to the reviewers' comments. Your revised manuscript is also likely to be sent to reviewers for further evaluation.

Sincerely,

Manuel Schibler

Associate Editor

Andrés Henao-Martínez

Deputy Editor

Reviewer's Responses to Questions

**Key Review Criteria Required for Acceptance?**

**Methods**

-Are the objectives of the study clearly articulated with a clear testable hypothesis stated?

-Is the study design appropriate to address the stated objectives?

-Is the population clearly described and appropriate for the hypothesis being tested?

-Is the sample size sufficient to ensure adequate power to address the hypothesis being tested?

-Were correct statistical analysis used to support conclusions?

-Are there concerns about ethical or regulatory requirements being met?

Reviewer #1: The objectives of this study was to analyze publication on LF treatment in the effort of finding better standards for LF studies concerning the inclusion/exclusion and outcome criteria as well as the clinical case definition. The study design was very robust concerning the search criteria as well as the analysis of the data. The number of included studies in this analysis is very comprehensive. The statistics are exclusively descriptive, as this study does not formulate a clasical hypothesis. There are no ethical concerns.

Reviewer #2: The study is a systematic review of the literature reporting the clinical signs and symptoms of Lassa fever. The suggested major revisions prior to acceptance are listed below.

1. The link between this review manuscript and how its data might be used to guide clinical trials needs to be better established. In the Introduction section, consider describing how other review papers guided clinical trials for treatments for other febrile illnesses. 

2. In the Discussion section consider adding a subsection describing how the results might be used to guide future ribavirin trials. Is there a proposed set of symptoms that should be considered for a suspected case definition?

3. While the focus of the study is to provide information regarding ribavirin trials, there is little information in the manuscript about selected studies that included ribavirin components. How many of the studies included ribavirin? Was ribavirin generally found to be effective in those studies? 

4. The premise of this article seems to be that Lassa symptoms can be standardized across geographic regions. What are the implications of standardization? Different countries have different strains of LF. Also, public health units differ by time to clinical presentation and laboratory capacity (for evaluating lab-based items contributing to a suspected case definition). While it may be beyond the scope of this study to report in detail on those challenges, they do seem worth mentioning. Consider adding a subsection in the Discussion section about the challenges of standardization. 

5. Consider adding a subsection in the Discussion sections about the study limitations; e.g., test validity, little distinction between acute and long-term exposure. For instance, IgG testing measures long-term exposure and is associated with different symptoms than Ag+ acute cases.

Reviewer #3: (No Response)

**Results**

-Does the analysis presented match the analysis plan?

-Are the results clearly and completely presented?

-Are the figures (Tables, Images) of sufficient quality for clarity?

Reviewer #1: The analysis follows a clear plan, the presentation of the result is clear and comprehensive. All figures and tables are of good quality. It is of note that the results include a broad range of clinical symptoms that go beyond what is often included in clincal case definitions. This corresponds to the numerous case reports that represent the broad clinical picture of LF and is thus an important contribution. The authors also recorded which clinical symptoms were more frequent after baseline. 

The authors alos analyzed the frequency of bleeding at different sites and found haematuria the most frequent presentation. However, it should be more thoroughly explained which kinds of haemturia were documented, and if cases of microhaematuria were included, as it can be a resut of kidney injury but does not necessarily represent true bleeding in the stricter sense to my opinion.

Overall, the results are very informative in the effort of finding better standards for clinical studies on LF

Reviewer #2: The analysis does match the analysis plan, the results are clearly and completely presented, and the figures are of sufficient quality for clarity.

Reviewer #3: (No Response)

**Conclusions**

-Are the conclusions supported by the data presented?

-Are the limitations of analysis clearly described?

-Do the authors discuss how these data can be helpful to advance our understanding of the topic under study?

-Is public health relevance addressed?

Reviewer #1: The conclusion are sound in nature but it should be pointed out a little more that this analysis is strictly descariptive and that in order to reach the declared goal of establishing standardized protocols, results would need to be graded (for discussion). the rest of the discussion of the result is very well suited to inform the understanding of the topic.

Reviewer #2: The link between this review manuscript and how its data might be used to guide clinical trials needs to be better established. In the Introduction section, consider describing how other review papers guided clinical trials for treatments for other febrile illnesses.

Reviewer #3: (No Response)

**Editorial and Data Presentation Modifications?**

Reviewer #1: Figure 3 needs to be graphically edited as the numbers are very hard to read, even on a large screen

Other than this minior issue I would recommend to accept

Reviewer #2: Minor suggested revisions:

1. Lines 12-13: "all studies were included"

This statement is overly broad as the literature search did not cover all available bibliographic databases and search engines (e.g., Google Scholar)

2. Line 17: what arose from these discussions?

3. Lines 54-56. Further evidence in the form of clinical trials is required both to confirm the efficacy and safety of ribavirin and to test new therapies.

While this statement is true, consider elaborating on this statement by mentioning the advantages of testing in additional geographic regions, larger sample sizes, accounting for different LF strains, etc.

4. Line 59-60: the clinical characteristics of LF and patient outcomes

The second component of the sentence clinical characteristics of patient outcomes is confusing. Consider rewording.

5. Line 87: spell out acronym for REDCap on first occurrence

6. Lines 102-113: Overuse of “We”

7. Line 225 (Table 5). The definition of laboratory confirmation is extremely influential, particularly at baseline. For instance, Ag ELISA measures acute exposure, IgM measures recent exposure, and IgG measures long-term exposure. It may be helpful to review Table 5 by diagnostic approach to determine whether pooling is justified.

8. Line 262 – “An agreed”

Consider replacing agreed with uniform.

9. Lines 284-287: What are the reasons for this? Limited laboratory capacity?

10. Discussion section: seems to be missing any mention of the need for improved diagnostics and rapid testing

Reviewer #3: (No Response)

**Summary and General Comments**

Reviewer #1: this paper provides a meaningful contribution and is the largest analysis of the kind to my knowledge.

Reviewer #2: General comments

The study addresses a void in the literature by providing a comprehensive systematic review of the literature reporting the clinical signs and symptoms of Lassa fever to guide future ribavirin trials.

Major comments

1. The link between this review manuscript and how its data might be used to guide clinical trials needs to be better established. In the Introduction section, consider describing how other review papers guided clinical trials for treatments for other febrile illnesses. 

2. In the Discussion section consider adding a subsection describing how the results might be used to guide future ribavirin trials. Is there a proposed set of symptoms that should be considered for a suspected case definition?

3. While the focus of the study is to provide information regarding ribavirin trials, there is little information in the manuscript about selected studies that included ribavirin components. How many of the studies included ribavirin? Was ribavirin generally found to be effective in those studies? 

4. The premise of this article seems to be that Lassa symptoms can be standardized across geographic regions. What are the implications of standardization? Different countries have different strains of LF. Also, public health units differ by time to clinical presentation and laboratory capacity (for evaluating lab-based items contributing to a suspected case definition). While it may be beyond the scope of this study to report in detail on those challenges, they do seem worth mentioning. Consider adding a subsection in the Discussion section about the challenges of standardization. 

5. Consider adding a subsection in the Discussion sections about the study limitations; e.g., test validity, little distinction between acute and long-term exposure. For instance, IgG testing measures long-term exposure and is associated with different symptoms than Ag+ acute cases.

Minor comments

1. Lines 12-13: "all studies were included"

This statement is overly broad as the literature search did not cover all available bibliographic databases and search engines (e.g., Google Scholar)

2. Line 17: what arose from these discussions?

3. Lines 54-56. Further evidence in the form of clinical trials is required both to confirm the efficacy and safety of ribavirin and to test new therapies.

While this statement is true, consider elaborating on this statement by mentioning the advantages of testing in additional geographic regions, larger sample sizes, accounting for different LF strains, etc.

4. Line 59-60: the clinical characteristics of LF and patient outcomes

The second component of the sentence clinical characteristics of patient outcomes is confusing. Consider rewording.

5. Line 87: spell out acronym for REDCap on first occurrence

6. Lines 102-113: Overuse of “We”

7. Line 225 (Table 5). The definition of laboratory confirmation is extremely influential, particularly at baseline. For instance, Ag ELISA measures acute exposure, IgM measures recent exposure, and IgG measures long-term exposure. It may be helpful to review Table 5 by diagnostic approach to determine whether pooling is justified.

8. Line 262 – “An agreed”

Consider replacing agreed with uniform.

9. Lines 284-287: What are the reasons for this? Limited laboratory capacity?

10. Discussion section: seems to be missing any mention of the need for improved diagnostics and rapid testing.

Reviewer #3: Merson et al., report valuable information about the signs, symptoms and outcomes of infection with Lassa virus. I believe that the following points are important and deserve further clarification.

More data regarding Lassa virus epidemiology in the 'Introduction' section may further enhance the structural feature of the manuscript. The article " Systematic review and meta-analysis of the epidemiology of Lassa virus in humans, rodents and other mammals in sub-Saharan Africa. PLoS Negl Trop Dis. 2020 Aug 26;14(8):e0008589" may be useful to provide such information to the readers.

Line 61: I would suggest that the authors reword the purpose of the study. “individual-level data” is confusing with the reviews performed on individual patient data from the original databases of the authors of the included studies. This concept of individual data from included study participants should also be clarified throughout the manuscript.

Line 99: The authors should clarify the detection methods and the targets (RNA, viral antigen, antibodies, IgM, IgG…) searched to be considered as confirmed cases of Lassa virus infection.

Line 102-106: In addition to the proportions reported, authors should ideally (if possible) provide the individual data of the included studies.

The signs and symptoms reported in this work should be fully discussed in relation to the multiple other febrile illnesses and haemorrhagic fevers due to other pathogens present in these endemic areas of West Africa and presenting the same clinical picture.

The discussion needs more references to better inform the concept of the research. For example, line 291, what is the work that argues that pregnant women are at higher risk of death from infection with Lassa virus.

The review authors should clearly describe the limitations of this article.

PLOS authors have the option to publish the peer review history of their article (what does this mean?). If published, this will include your full peer review and any attached files.

Reviewer #1: No

Reviewer #2: Yes: Jeffrey G. Shaffer

Reviewer #3: No
---

## [Decision Letter · Decision Letter 1]

3 Sep 2021

Dear Mrs. Bourner,

We are pleased to inform you that your manuscript 'Clinical characterization of Lassa fever: a systematic review of clinical reports and research to inform clinical trial design' has been provisionally accepted for publication in PLOS Neglected Tropical Diseases.

**Before your manuscript can be formally accepted, we would like you to address two more minor concerns:**

**1. Page 5, line 54: please replace "although this figure is lower (12%) in more recent studies" by "although this figure is lower (12%) in a more recent study"**

**2. Page 9, lines 150-151: please correct the percentage of quasi-randomised studies; it's 1 (or 1.4), not 2%**

You will also need to complete some formatting changes, which you will receive in a follow up email. A member of our team will be in touch with a set of requests.

Best regards,

Manuel Schibler

Associate Editor

Andrés Henao-Martínez

Deputy Editor

Reviewer's Responses to Questions

**Key Review Criteria Required for Acceptance?**

**Methods**

-Are the objectives of the study clearly articulated with a clear testable hypothesis stated?

-Is the study design appropriate to address the stated objectives?

-Is the population clearly described and appropriate for the hypothesis being tested?

-Is the sample size sufficient to ensure adequate power to address the hypothesis being tested?

-Were correct statistical analysis used to support conclusions?

-Are there concerns about ethical or regulatory requirements being met?

Reviewer #3: (No Response)

**Results**

-Does the analysis presented match the analysis plan?

-Are the results clearly and completely presented?

-Are the figures (Tables, Images) of sufficient quality for clarity?

Reviewer #3: (No Response)

**Conclusions**

-Are the conclusions supported by the data presented?

-Are the limitations of analysis clearly described?

-Do the authors discuss how these data can be helpful to advance our understanding of the topic under study?

-Is public health relevance addressed?

Reviewer #3: (No Response)

**Editorial and Data Presentation Modifications?**

Reviewer #3: (No Response)

**Summary and General Comments**

Reviewer #3: Although the authors have adequately addressed my suggestions, I think that the citation of the 30% case fatality ratio from the meta-analysis [5] might not be attenuated by the citation [6] which is an original article and therefore difficult to compare. In addition, the authors of the present review report in the abstract the same 30% case fatality ratio found from their own work.

PLOS authors have the option to publish the peer review history of their article (what does this mean?). If published, this will include your full peer review and any attached files.

Reviewer #3: No

---

## [Editor Report · Acceptance letter]

15 Sep 2021

Dear Ms Bourner,

We are delighted to inform you that your manuscript, "Clinical characterization of Lassa fever: a systematic review of clinical reports and research to inform clinical trial design," has been formally accepted for publication in PLOS Neglected Tropical Diseases.

Best regards,

Shaden Kamhawi

co-Editor-in-Chief

Paul Brindley

co-Editor-in-Chief
